# Horizontal Federated Heterogeneous Graph Learning: A Multi-Scale Adaptive Solution to Data Distribution Challenges

## Abstract

Federated heterogeneous graph learning, an extension of federated learning, enables effective representation of complex multidimensional relationships while preserving data privacy. In horizontal federated heterogeneous graph learning, data from different parties often differ in topology and semantic distributions, causing sensitivity to distribution imbalance and amplifying the complexity of the topological structure. This interaction makes it difficult for models to learn shared representations, leading to increased instability during training. To address these challenges, this paper proposes a novel multi-scale adaptive horizontal federated heterogeneous graph learning method MAFedHGL. A random masking mechanism forces the model to infer missing connections. The model also captures multi-hop and multi-path connections using high-order topology mining, enhancing robustness against structural heterogeneity. Dynamic semantic consistency modeling uses a masking matrix to recover and integrate diverse node attributes, ensuring both global and local semantic consistency. Using clustering coefficients as aggregation weights enables clients with richer structural information to contribute more effectively to the global model, improving adaptability and performance across varying data distributions in horizontal federated heterogeneous graph learning. Extensive experiments on multiple public heterogeneous graph datasets validate that the proposed method outperforms state-of-the-art methods in both performance and robustness across various data distribution scenarios.

## CCS Concepts

• **Computing methodologies** → **Distributed artificial intelligence**; • **Security and privacy** → *Distributed systems security*; • **Networks** → *Network privacy and anonymity*.

## Keywords

federated heterogeneous graph learning, federated learning, heterogeneous information network

**ACM Reference Format:**
Anonymous Author(s). 2024. Horizontal Federated Heterogeneous Graph Learning: A Multi-Scale Adaptive Solution to Data Distribution Challenges. In . ACM, New York, NY, USA, 10 pages. https://doi.org/10.1145/nnnnnnn.nnnnnnn

## 1 INTRODUCTION

In real-world applications, heterogeneous graphs (HGs) are often derived from multiple data sources, involving sensitive information such as users' social connections, transaction data, or academic collaboration networks [43, 44]. Privacy and security concerns make traditional centralized training models difficult to apply directly. Federated learning (FL) solves this issue by allowing participants to train models locally, sharing only model parameters instead of raw data [2, 27]. By overcoming the limitations of data silos, FL improves the training effectiveness of heterogeneous graph neural network (HGNN). This feature is crucial in domains such as healthcare, finance, and social networks, where sensitive data is prevalent. FL preserves privacy while leveraging diverse data sources, leading to better model performance [18, 22]. Integrating FL with HGNN allows exploitation of heterogeneous data richness while complying with strict privacy regulations, advancing HGNN applications in privacy-sensitive fields [6, 23].

Federated heterogeneous graph learning (FHGL) combines FL and HGNN to address HG data distribution across participants. Each participant trains a model locally on its own HG data, transmitting the model parameters to a central server for aggregation through the FL framework. FHGL is well-suited for scenarios with distributed HG data. For instance, banks and financial institutions can enhance the performance of models for financial risk management and fraud detection through local training, all while safeguarding user privacy.

In a FL environment, the non-independent and identically distributed (non-IID) nature of data exacerbates the challenges of model robustness and adaptability, particularly in cases of imbalanced feature and label distributions [1, 20]. Compared to traditional Euclidean data, such as images or text, training HGNN in FL settings faces more complex data heterogeneity issues [9, 21, 26]. Horizontal federated heterogeneous graph learning (HFHGL) is a special case of FHGL where participants possess HG structures of similar types, with nodes and edges sharing the same schema, but each holding distinct data samples. The distribution characteristics of heterogeneous data further amplify the difficulty of HFHGL, that intensify the challenges introduced below.

First, the FL framework magnifies the structural sensitivity of HGNN [6, 13, 46]. Clients may hold significantly different types and numbers of nodes and edges, leading to diverse local topological structures across clients. HGNN is highly sensitive to graph structure, and this sensitivity is even more pronounced in heterogeneous graphs, where varying topological information can drastically affect the learning outcomes. When there are large structural differences between clients' local graphs, the global model may struggle to capture the overall characteristics of the heterogeneous graph, making it difficult to generalize effectively across all clients. Secondly, semantic complexity is another critical challenge in training HGNN [29, 50]. Heterogeneous graphs are composed of multiple types of

nodes and edges, and the semantics of each node type depend on the relationships and types of neighboring nodes. These neighborhood relationships often vary significantly between clients. This relationship dependency makes it difficult to unify the embedding representations across clients. In the FL setting, where clients hold data with varying semantic information, the global model may struggle to align these diverse semantic patterns. The discrepancy in the semantic information captured by different clients can hinder the global model's ability to effectively aggregate and generalize across all participants, further complicating the training process. The heterogeneity in both structural and semantic aspects of the data in FL scenarios poses unique challenges for HGNN, making it more difficult to achieve optimal performance. Moreover, traditional FL often adopts simple averaging aggregation methods, which assume that the data across clients is homogeneous and of similar importance [36]. However, in the context of FHGL, average aggregation strategies fail to distinguish between the quality and structural complexity of data from different clients. The graph structural characteristics of different clients in heterogeneous graphs may vary significantly, and simple averaging can result in the loss of critical information from some local graphs. In extreme cases, some clients may have poor graph structures, and including these clients in average aggregation can introduce unnecessary noise into the global model.

To address the challenges in HFHGL, we propose a novel **M**ulti-scale **A**daptive Horizontal **Fed**erated **H**eterogeneous **G**raph **L**earning method MAFedHGL. Through High-order Topology Mining (HTM), the model captures multi-hop and multi-path connections between nodes, enabling comprehensive analysis of multi-scale features in complex graph structures. During the training process, a local structure-based random masking mechanism is employed, where portions of the edge information are hidden, forcing the model to infer the missing connections from the remaining structure. This design compels the model to focus on long-range correlations between distant nodes, supplementing the missing information with indirect connections, such as multi-hop paths or more intricate topological patterns representing high-order structures. By learning these high-order structures, the model effectively uncovers the latent topological relationships across different scales, enhancing the robustness of HFHGL in dealing with structural heterogeneity across different clients. In terms of semantic modeling, we address missing semantic relationships by fully utilizing the complex semantic information inherent in heterogeneous graphs, achieving adaptive Semantic Consistency Modeling (SCM). A dynamic masking matrix is used to obscure node semantic attributes, with the HGNN recovering the missing information. This approach helps the model aggregate the attributes of different types of nodes, integrating complicated topological structures to ensure both global and local semantic consistency. This process not only enhances the feature representation of local nodes but also strengthens the model's global semantic understanding by integrating multi-level semantic relationships, making it more resilient in distributed HG environments. Additionally, we introduce clustering coefficients as aggregation weights for clients, measuring the tightness of node-to-neighbor connections. In this way, during federated aggregation, clients with more complex structural information contribute more to the global model, allowing the model to dynamically adapt to varying data distribution scenarios and improving the overall performance and robustness of the global model. With this design, the model demonstrates stronger generalization ability and stability when addressing the issue of distribution heterogeneity in HFHGL.

In summary, the main contributions of this paper are as follows:

(1) To the best of our knowledge, this is the first study on horizontal federated heterogeneous graph learning, which is of significant practical value in real-word. We further formalize the concept of horizontal federated heterogeneous graph learning.
(2) We construct high-order topology mining and semantic consistency modeling to address the issue of poor model robustness caused by imbalanced client data distribution and large graph structural differences in a multi-scale and adaptive way. We further propose a federated heterogeneous graph aggregation strategy, assigning higher weights to clients with tightly connected structures, thus enhancing the aggregation effect.
(3) Experiments on four public heterogeneous graph datasets, with tailored partitioning strategies for horizontal federated learning, demonstrate that our method consistently achieves high model performance across various data distributions while maintaining robustness across multiple trials.

## 2  RELATED WORKS

### 2.1  Heterogeneous Graph Neural Network

HGNN are essential for processing graph data with complex structures, involving various types of nodes and edges [32]. In recent years, research progress in HGNN has primarily focused on capturing complex structures, handling heterogeneous information, and improving model robustness. Hin2Vec [5] propose a metapath-based heterogeneous information network embedding model that learns embeddings by exploiting the relationships between different types of nodes. Similarly, MAGNN [7] further capture the composite relationships in HG by defining multiple metapaths to achieve node embeddings. To address the structural challenges of HG, methods like [14, 47] use graph transformation networks to learn effective node representations based on new graph structures. Some studies [11, 40, 48, 52] use hierarchical attention mechanisms at node and semantic levels to effectively model the importance of metapaths and node relationships. Additionally, frameworks such as [12, 30, 39, 53] leverage generative adversarial networks (GANs) to enhance the robustness of heterogeneous graph embeddings. HGMAE [37] introduces dynamic masking strategies in self-supervised learning, bypassing the complexity of negative sampling in traditional contrastive learning, thus improving robustness and generalization. Although these methods have made significant strides in processing HG's structural and semantic information, they still struggle with distributed data handling and privacy protection. Most current methods depend on centralized data processing, resulting in data heterogeneity, low communication efficiency, and privacy risks in distributed environments.

## 2.2 Federated Learning

As FL continues to evolve, recent research has introduced various methods to tackle challenges posed by data heterogeneity and non-IID data. Some studies propose contrastive learning techniques to narrow the gap between local and global models, addressing local update drift due to data distribution differences [3, 8, 17]. MOON [19] notably improves model representation consistency, leading to better convergence in non-IID data settings. Other studies incorporate knowledge distillation techniques [33, 45, 54] to mitigate performance degradation that can occur with direct model aggregation. For instance, FedFTG [51] utilizes data-free knowledge distillation by leveraging a generator on the server side to fine-tune the global model. Methods like [1, 9, 16, 20, 26] reduce variance in local updates by introducing control variables and proximal terms, alleviating the negative impact of heterogeneous data on federated learning convergence. Research on data prototypes [4, 25, 28, 49] shows that it helps alleviate performance degradation caused by data heterogeneity and enhances personalized learning. FedPROTO [35] propose a prototype-based federated learning approach, which aggregates class prototypes from clients to form a global prototype, guiding local learning. Additionally, robust aggregation algorithms [22, 41] enhance model aggregation by controlling aggregation weights. For example, FedAW [36] achieves better performance and convergence by adjusting aggregation weights based on the quality of local data. Although these methods tackle data heterogeneity in model training, they struggle with non-Euclidean data, especially graph data. The complex topology and node relationships in graphs cause local model updates to drift more easily, making it harder for the global model to capture overall data characteristics.

## 2.3 Federated Graph Learning

Research on federated graph learning (FGL) begin as a response to privacy concerns when training graph neural network (GNN) across multiple distributed clients. It focuses on collaborative training by sharing model parameters instead of data [24, 29, 34]. Early studies primarily focus on homogeneous graph neural networks. Some methods [10, 31, 42, 46] achieve semi-supervised node classification while preserving data privacy, balancing communication overhead and model convergence speed. As research progressed, more works begin to focus on FL on subgraph data. FedSAGE [50] compensates for missing links between subgraphs by generating missing neighbors, enhancing the model's generalization capabilities. To tackle data heterogeneity, researchers propose FedLit [43], which handles link heterogeneity in graphs through an edge clustering module and a multi-channel graph convolution network. Subsequently, FL has gradually expanded into the domain of HGNN. HG, compared to homogeneous graphs, have more complex node and edge types, posing greater challenges in FL scenarios. FedHRec [44] aims to address privacy-preserving recommendation problems by using a FL framework that keeps users' private data on clients while sharing part of the heterogeneous information network on the server. FedHGN [6] introduced mode weight decoupling and coefficient alignment to optimize the FL process on HG, where the schema of HG differs across clients. Although scenarios where clients possess different schemas present a complex and worthwhile research challenge, in real-world applications, uniform client schemas often

do not have significant privacy concerns. For example, in collaborations between different banks, it is relatively straightforward to align the fields they collect in user records. However, the specific fields associated with individual users, which may differ across banks, often entail higher privacy requirements and distributional variance.

## 3 PRELIMINARY

This section will introduce the fundamental concepts involved in this paper.

DEFINITION 1. **Heterogeneous graph.** A heterogeneous graph (HG) $\mathcal{G}$ is defined as a graph with multiple types of nodes and edges, represented as: $\mathcal{G} = (\mathcal{V}, \mathcal{E}, \mathcal{A}_V, \mathcal{A}_E)$ where $\mathcal{A}_V : \mathcal{V} \to \mathcal{T}_V$ and $\mathcal{A}_E : \mathcal{E} \to \mathcal{T}_E$ map each node/edge to a specific type in set $\mathcal{T}_V$ and $\mathcal{T}_E$

DEFINITION 2. **Meta-path.** A meta-path $P$ is a sequence of node types $T$ and edge types $R$, formally represented as: $P = (T_1 \xrightarrow{R_1} T_2 \xrightarrow{R_2} \dots \xrightarrow{R_{n-1}} T_n)$, where $T_i \in \mathcal{T}_V$ and $R_i \in \mathcal{T}_E$.

DEFINITION 3. **Heterogeneous Graph Neural Network.** A heterogeneous graph neural network (HGNN) is a neural network designed to handle HG, where nodes and edges belong to different types. The objective of a HGNN is to learn node embeddings $\mathbf{H}_v \in \mathbb{R}^d$, $d$ is the dimension of the embedding. The network is built based on a message-passing mechanism which updates node representations $\mathbf{H}_v^{(l)}$ for each node $v$ at layer $l$ by aggregating information from their neighbors $\mathcal{N}_v^{(l)}$.

$$\mathbf{m}_v^{(l)} = AGGREGATE_{u \in \mathcal{N}_v^{(l)}}^{(l)} \left( \left\{ f_{\mathcal{A}_E(e_{vu})} \left( \mathbf{H}_u^{(l)}, \mathbf{H}_v^{(l)}, \mathbf{W}_{\mathcal{A}_E(e_{vu})}^{(l)} \right) \right\} \right),$$ (1)

$$\mathbf{H}_v^{(l+1)} = UPDATE^{(l)} \left( \mathbf{H}_v^{(l)}, \mathbf{m}_v^{(l)} \right),$$ (2)

where $f_{\mathcal{A}_E(e_{vu})}$ is a message-passing function that depends on the type of the edge $e_{vu}$, $\mathbf{W}_{\mathcal{A}_E(e_{vu})}^{(l)}$ is a type-specific weight matrix. Then, relationships between different types of nodes can be captured through meta-paths $P$, where $\mathbf{W}_P^{(l)}$ is the weight matrix for meta-path $P$.

$$\mathbf{H}_v^{(P,l+1)} = AGGREGATE_P^{(l)} \left( \left\{ f_P \left( \mathbf{H}_u^{(l)}, \mathbf{H}_v^{(l)}, \mathbf{W}_P^{(l)} \right) \mid u \in \mathcal{N}_v^P \right\} \right),$$ (3)

To combine multiple meta-paths, the representations from different meta-paths can be combined using a weighted sum or an attention mechanism, where $\sigma_P$ is the attention weight for meta-path $P$.

$$\mathbf{H}_v^{(l+1)} = \sum_{P \in \mathcal{P}} \sigma_P \mathbf{H}_v^{(P,l+1)},$$ (4)

DEFINITION 4. **Federated Heterogeneous Graph Learning.** Federated heterogeneous graph learning (FHGL) is a distributed machine learning approach that enables $K$ participants (e.g., devices or servers) to collaboratively train a global HGNN $w$ without sharing their local HG. The goal of a FHGL is to minimize the following objective function:

$$\min_w F(w) = \sum_{k=1}^{K} \alpha_k F_k(w),$$ (5)

where $F(w)$ is the global objective function, representing global HGNN's performance across all participants' HG. $F_k(w)$ is the local objective

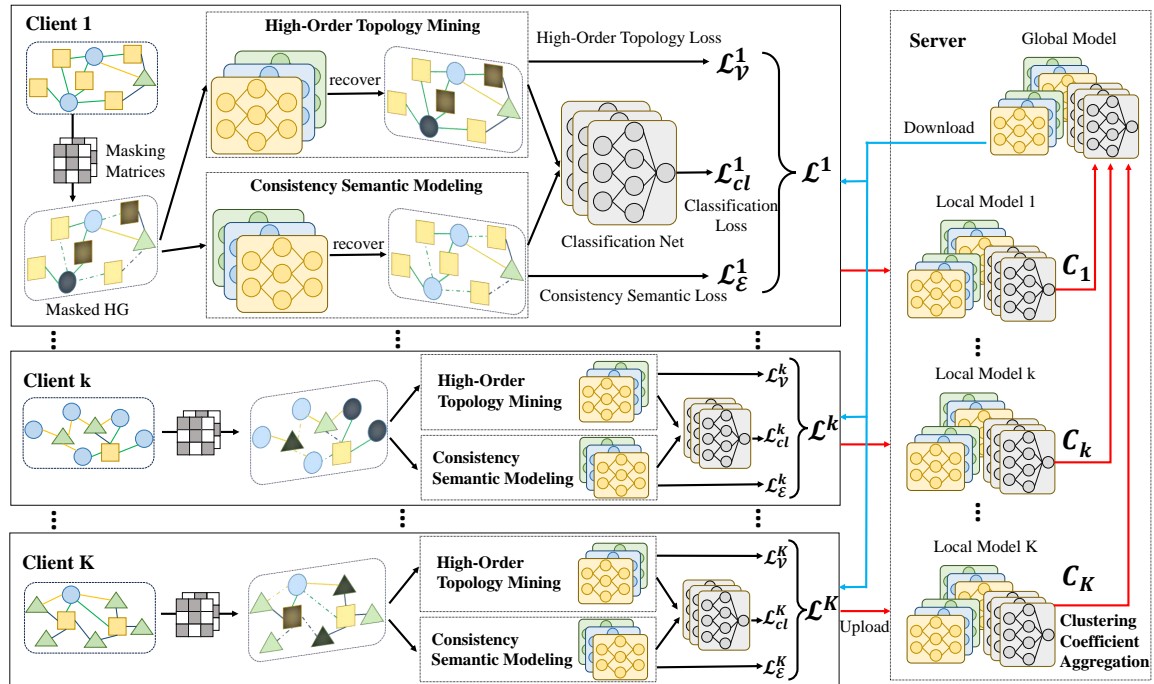

**Figure 1: Framework of proposed MAFedHGL. The key methods of MAFedHGL include high-order topological structure mining, consistent semantic modeling, and a clustering coefficient-based aggregation strategy. HTM enhances the model's perception of multi-hop and indirect connections, allowing it to effectively capture global structural information of the graph. Through CSM, MAFedHGL learns to recover missing information by dynamically masking partial node attributes, ensuring semantic consistency across clients. What's more, the clustering coefficient-based federated aggregation strategy adjusts the aggregation weights according to the structural tightness of each client's graph, enabling the global model to focus more on valuable graph structures.**

*function of participant $k$, representing the HGNN's performance on that participant's local HG. $\alpha_k$ is the aggregate weight for client $k$.*

Based on the above definitions, we then present our task of this paper:

For a FHGL system with $K$ clients, each client holds a dataset $D^k = \{\mathcal{G}^k, \mathcal{X}^k, \mathcal{Y}^k, \mathcal{A}_V^k, \mathcal{A}_E^k\}$, where $\mathcal{X}^k$ represents the local node feature set, denoted as $\mathcal{X}^k = \{X_{T_1}^k, X_{T_2}^k, ..., X_{T_{|\mathcal{A}_V|}}^k\}$, $\mathcal{Y}^k$ represents the corresponding label features, as $\mathcal{Y}^k = \{Y_{T_1}^k, Y_{T_2}^k, ..., Y_{T_{|\mathcal{A}_V|}}^k\}$, and $\mathcal{G}^k = (\mathcal{V}^k, \mathcal{E}^k, \mathcal{A}_V, \mathcal{A}_E)$. Specifically, we denote the node features $X_{T_{target}}^k$ and labels $Y_{T_{target}}^k$ corresponding to the target node type $T_{target}$ to be classified as $X$ and $Y$ respectively. The learning objective is to classify the nodes based on their features, in other word, to find the optimal function $F(w)$. On each client, the local function $F_k(w)$ can be represented as:

$$F_k(w) = \mathcal{L}_{cl}^k = -\sum_{v \in \mathcal{V}_{train}^k} Y_v \log \hat{Y}_v, \qquad (6)$$

DEFINITION 5. ***Horizontal Federated Heterogeneous Graph Learning.*** *Horizontal federated heterogeneous graph learning (FHGL) is a specific category of FHGL, where participants share the same types of HG structures, which means $\mathcal{A}_V^k = \mathcal{A}_V, \mathcal{A}_E^k = \mathcal{A}_E, \mathcal{T}_V^k =$*

*$\mathcal{T}_V, \mathcal{T}_E^k = \mathcal{T}_E$, but the nodes and edges contained in each participant's graph are not identical, particularly with regard to the target samples for classification in federated heterogeneous graph node classification task.*

## 4 METHODOLOGY

In this section, we provide a detailed explanation of the implementation of MAFedHGL. Specifically, MAFedHGL introduces mask-based high-order topology mining and consistency semantic modeling to assist clients in learning HG node embeddings. Subsequently, aggregation weights are assigned to each client based on the clustering coefficient of its local graph, improving the stability of the aggregation process. The framework of MAFedHGL is illustrated in Figure 1.

### 4.1 Higher-order Topology Mining

In HG, relationships between nodes are not solely determined by direct neighbor connections; rather, complex higher-order topological relationships exist. These higher-order relationships refer to the interactions between nodes formed through multi-hop, multi-path, or indirect connections. By mining higher-order topological structures, a model can effectively capture these multi-scale graph

structure features, thereby enhancing its understanding of the overall graph morphology.

In local HGNN training, information of edges in the input graph are selectively hidden at random. This can be represented as

$$\mathbf{A}_{masked} = \mathbf{M}_E \odot \mathbf{A}, \tag{7}$$

where $\mathbf{M}_E$ is the masking matrix, and $\odot$ denotes element-wise multiplication. The model then performs hierarchical learning through a heterogeneous graph convolutional network, where each layer captures information from local neighborhoods to higher-order topological structures

$$\mathbf{H}^{(L)} = \sigma \left( \tilde{\mathbf{D}}_{masked}^{-\frac{1}{2}} \tilde{\mathbf{A}}_{masked} \tilde{\mathbf{D}}_{masked}^{-\frac{1}{2}} \mathbf{H}^{(L-1)} \mathbf{W}^{(L-1)} \right), \tag{8}$$

where $\tilde{\mathbf{A}}_{masked} = \mathbf{A}_{masked} + \mathbf{I}$ and $\tilde{\mathbf{D}}_{masked}$ is the degree matrix corresponding to the adjacency matrix $\tilde{\mathbf{A}}_{masked}$. During the hierarchical learning process, the convolution operation is repeatedly stacked, allowing node features to propagate from adjacent nodes to more distant ones. This mechanism forces the model not only to rely on local direct connection information but also to infer the hidden features from the remaining graph structure. The hidden information of edge relationship information drives the model to learn how to capture topological information from distant nodes or higher-order graph paths.

The model needs to learn the similarity between nodes under a given meta-path and use this similarity to determine whether a higher-order topological relationship exists. Given two nodes $v$ and $u$, connected through a meta-path $p$, the model needs to predict whether there is relationship between $v$ and $u$ by:

$$\hat{A}_{vu}^P = f(h_v^P, h_u^P), \tag{9}$$

where $\hat{A}_{vu}^P$ represents the higher-order topological index of nodes $v$ and $u$ based on meta-path $p$, and $f(\cdot)$ is the cosine similarity function.

To extract higher-order topological information from the meta-path, the model aggregates different types of node features along meta-path $p$. The feature representation of node $v$ under meta-path $p$, denoted as $h_v^P$, can be expressed as

$$h_v^P = \sum_{u \in \mathcal{N}_v^P} \eta_{uv}^P h_u, \tag{10}$$

where $\eta_{uv}^P$ is the attention weight between $v$ and $u$ under meta-path $P$.

To optimize the accuracy of higher-order topology mining, the loss is computed based on edge reconstruction. The model minimizes the following loss function:

$$\mathcal{L}_{\mathcal{E}} = - \sum_{(v,u) \in \mathcal{V}} [A_{vu} log \hat{A}_{vu}^P + (1 - A_{vu}) log(1 - \hat{A}_{vu}^P)], \tag{11}$$

By learning higher-order topological structures, the model can understand not only local node relationships but also capture the global distribution of the graph. This enables the model to form a globally consistent understanding of the entire graph when dealing with complex heterogeneous graphs, thus enhancing its reasoning and generalization capabilities.

## 4.2 Consistency Semantic Modeling

From the perspective of consistency semantic modeling (CSM), the model focuses on recovering masked node attribute information by predicting the masked target attributes. CSM aims to learn both global and local semantic information of nodes and establish a adaptive semantic representation of the nodes. The objective of CSM is to ensure that even in cases where partial node or edge information is missing, the model can recover the missing attributes by leveraging other visible information in the graph. This requires the model to not only comprehend the local node-edge structure but also to construct a consistent semantic representation in the global context.

Randomly masking parts of certain node attributes forces the model to rely on the unmasked attributes of other nodes, as well as the topological relationships between nodes, to infer the masked information. To further enhance the robustness and generalization ability of the model when handling HG, the model dynamically adjusts the masking ratio during training based on the complexity of different nodes or features, rather than using a fixed masking ratio. Thus, during the heterogeneous graph convolution, the node representation is updated as:

$$\mathbf{H}_0 = \mathbf{M}_N \mathbf{X}, \tag{12}$$

where $\mathbf{M}_N = \varphi \cdot exp^{-\lambda r}$, $\varphi$ is the initial masking rate, $\lambda$ is the rate controlling how the masking ratio changes over the training epochs, and $r$ is the current global training epoch.

At the early stages of training, the model applies a higher masking rate that more features are masked, forcing the model to learn from a limited amount of input features or neighborhood information. As training progresses, the dynamic masking rate gradually decreases, allowing the model to utilize more node features, thereby improving its accuracy in reconstruction tasks. In this way, the model progressively transitions to handling scenarios with less masking, enhancing its adaptability to complex graph structures.

To ensure the randomness of information masking and the stability of model convergence, before applying feature masking, a portion of node samples will retain their features unaffected with a certain probability $\phi_{un}$. Additionally, with a certain probability $\phi_{re}$, some sample features will be replaced by the unmasked features of neighboring nodes.

Once the node representations are updated through the aggregation operation, the model attempts to recover the masked target attributes. The recovered attributes can be represented as:

$$\hat{x}_v = g(h_v^{(L)}), \tag{13}$$

where $\hat{x}_v$ is the target feature of node $h_v^{(L)}$ is the final representation generated through multi-layer aggregation, and $g(\cdot)$ is the decoding function responsible for mapping the high-dimensional node representation back to the attribute space.

To ensure that the recovered attributes closely match the true attributes, the model adopts an attribute reconstruction loss function to measure the discrepancy between the recovered values and the ground truth:

$$\mathcal{L}_{\mathcal{V}} = \frac{1}{|\mathcal{V}_{masked}|} \sum_{v \in \mathcal{V}_{masked}} ||x_v - \hat{x}_v||_2^2, \tag{14}$$

where $\mathcal{V}_{masked}$ represents the set of masked nodes. This loss function enables the model to minimize the difference between the predicted and true attributes, thus improving the accuracy of the recovery process.

The ultimate learning objective $\mathcal{L}^k$ on the client $k$ can be defined as the weighted sum of the high-order topology loss $\mathcal{L}^k_{\mathcal{E}}$, consistency semantic loss $\mathcal{L}^k_{\mathcal{V}}$, and classification loss $\mathcal{L}^k_{cl}$:

$$\mathcal{L}^k = \mathcal{L}^k_{cl} + \gamma_{\mathcal{E}} \mathcal{L}^k_{\mathcal{E}} + \gamma_{\mathcal{V}} \mathcal{L}^k_{\mathcal{V}}, \tag{15}$$

where $\gamma_{\mathcal{E}}$, $\gamma_{\mathcal{V}}$ are the aggregation weights corresponding to $\mathcal{L}^k_{\mathcal{E}}$, $\mathcal{L}^k_{\mathcal{V}}$ respectively.

### 4.3 Federated Clustering Coefficient Aggregation

In HG, the structure of different types of nodes and edges is complex, and in FL environments, the data distribution across clients can vary significantly. The clustering coefficient is a metric used to measure the tightness of connections between a node and its neighbors, reflecting local structural properties of the graph. Introducing clustering coefficient into FHGL can effectively capture the strength of connections between nodes within the local graph structures of each client.

We proposes a meta-path-based method, which involves defining a specific meta-path to calculate clustering coefficient $C_k$ for client $k$. Suppose we choose a specific meta-path $P$, we can compute the number of triangles and triples along this meta-path, and thereby calculate $C_k$, follow formula as:

$$C = \sum_P \sum_v \frac{|\{(u,w) : u, w \in \mathcal{N}_P(v), (u,w) \in \mathcal{E}_P\}|}{|\mathcal{N}_P(v)| \, (|\mathcal{N}_P(v)| - 1)}, \tag{16}$$

where $E_p$ represents the set of edges conforming to the meta-path $P$.

The global server collects $C_k$ from each client and calculates the aggregation weight of the client based on the number of nodes and the $C_k$ in the local graph. The aggregate weight for client $k$ is thus defined as:

$$\alpha_k = \frac{C_k \cdot |\mathcal{V}_k|}{\sum_k \{C_k \cdot |\mathcal{V}_k|\}}. \tag{17}$$

A higher $C_k$ in a client's local graph indicates more tightly connected nodes, which may contain more valuable structural information. By assigning higher aggregation weights to clients whose subgraphs exhibit higher $C_k$, the global model can be guided to focus more on these valuable subgraphs, allowing the model to learn useful features from the complex graph structures more effectively.

### 4.4 Privacy Analysis

In MAFedHGL, each client sends only the model weights to the central server, rather than the original HG data. The central server is responsible solely for aggregating these local models to generate a global model, significantly reducing the risk of data privacy breaches. Additionally, the server collects clustering coefficients from each client. However, relying solely on clustering coefficients and model parameters does not allow inference on the local data, thus satisfying the privacy requirements of FL. Although the designed local models attempt to reconstruct local data to learn latent features, both adjacency and node information are masked, and

node attributes are further perturbed through feature swapping. These techniques increase data obfuscation and incompleteness, reducing the dependency on real data during training and mitigating the potential risk of privacy leakage.

## 5 EXPERIEMENTS

### 5.1 Experimental Setup

**Datasets.** We conducted experiments on four publicly available HG datasets: **ACM** [47], **DBLP** [7], **IMDB** [47], and **Yelp** [12]. These datasets encompass various domains, including academia, movie reviews, and E-commerce, each with distinct structural and semantic characteristics. This diversity provides a comprehensive validation of our method's effectiveness and robustness across different application scenarios.

Table 1: The partitioning details of client datasets, showing the number of nodes $|\mathcal{V}_k|$, ratio between number of edges and nodes $|\mathcal{E}_k|/|\mathcal{V}_k|$, and $C_k$. It is evident that as the heterogeneity of the target node distribution increases, the disparity in data volume between clients becomes more pronounced. Notably, $C_k$ effectively evaluates the information density of heterogeneous graphs on each client under different levels of heterogeneity.

| Datasets | Partition | $|\mathcal{V}_k|$ | $|\mathcal{E}_k|/|\mathcal{V}_k|$ | $C_k$ |
|---|---|---|---|---|
| ACM | Uniform | 2201±112 | 2.350±0.081 | 1.95±0.54 |
| | $\beta$=10 | 2201±295 | 2.351±0.079 | 2.12±0.53 |
| | $\beta$=1 | 2117±841 | 2.382±0.176 | 2.87±0.95 |
| | $\beta$=0.1 | 1948±1415 | 2.321±0.362 | 4.30±2.15 |
| DBLP | Uniform | 4291±1557 | 1.742±0.253 | 0.07±0.06 |
| | $\beta$=10 | 4294±1717 | 1.735±0.253 | 0.08±0.07 |
| | $\beta$=1 | 4218±2162 | 1.760±0.236 | 0.11±0.10 |
| | $\beta$=0.1 | 3972±3750 | 1.802±0.240 | 0.16±0.15 |
| IMDB | Uniform | 2317±253 | 2.053±0.227 | 0.31±0.15 |
| | $\beta$=10 | 2307±366 | 2.055±0.233 | 0.34±0.23 |
| | $\beta$=1 | 2223±942 | 2.061±0.269 | 0.48±0.32 |
| | $\beta$=0.1 | 2025±1583 | 2.038±0.310 | 1.26±1.00 |
| Yelp | Uniform | 1637±234 | 9.053±3.855 | 18.95±9.50 |
| | $\beta$=10 | 1628±266 | 9.028±4.062 | 18.77±9.32 |
| | $\beta$=1 | 1538±497 | 8.850±4.570 | 20.18±10.75 |
| | $\beta$=0.1 | 1326±913 | 8.130±5.403 | 21.92±14.85 |

**Data Partitioning.** Our focus is on the scenario where clients share the same schema, but the distribution of nodes and edges varies. To construct a FL environment, we partitioned the HG datasets by simulating the methods used for Euclidean data in FL, distributing nodes and their corresponding edges across different clients. We adopt two data partitioning methods. In the `Uniform` approach, target nodes are evenly divided into $K$ subsets, and the subgraph corresponding to each subset is assigned to a client as a HG. In the Dirichlet approach, target nodes are partitioned using a `Dirichlet` distribution $D(\beta)$, also constructing subgraphs, which leads to significant differences in label distributions of target nodes across clients, thereby affecting the distribution of node types,

**Table 2: Accuracy results for 4 datasets with different partitioning methods, showing the mean and standard deviation of accuracy across different data partition scenarios on 4 datasets. A higher mean indicates better performance, while a lower standard deviation reflects greater stability across repeated experiments.**

| Dataset | Partition | FedAvg | FedProx | MOON | SCAFFOLD | FedDyn | FedAW | FedWT | FedPROTO | FedTGP | FedGCN | FedSAGE | FedLit | FedHGN | MAFedHGL |
|---|---|---|---|---|---|---|---|---|---|---|---|---|---|---|---|
| ACM | Uniform | 79.57±2.47 | 77.15±2.39 | 77.97±4.80 | 79.27±2.81 | 64.75±1.71 | 79.53±2.01 | 80.36±2.33 | 74.27±6.64 | 78.47±3.35 | 57.43±5.29 | 65.58±2.23 | 64.82±1.39 | 65.46±1.82 | **83.20±1.30** |
| | β=10 | 79.03±2.17 | 76.69±1.28 | 76.39±1.30 | 79.48±1.11 | 65.26±2.03 | 80.70±1.29 | 80.17±1.43 | 71.91±4.79 | 79.18±4.68 | 62.91±6.93 | 67.10±2.42 | 71.63±1.32 | 72.62±1.36 | **83.60±0.49** |
| | β=1 | 83.84±3.64 | 81.58±3.34 | 81.59±3.36 | 85.07±2.06 | 74.98±3.65 | 84.97±2.54 | 85.13±2.82 | 77.39±4.79 | 82.54±4.22 | 71.11±4.56 | 72.23±2.58 | 82.18±2.87 | 84.66±3.36 | **87.32±1.69** |
| | β=0.1 | 96.68±2.12 | 95.88±2.89 | 95.79±3.30 | 96.50±2.04 | 97.20±1.82 | 96.94±2.08 | 96.00±1.95 | 95.32±3.27 | 95.98±2.55 | 74.27±5.62 | 75.98±2.36 | 93.07±3.69 | 96.56±2.37 | **97.34±1.62** |
| DBLP | Uniform | 53.96±1.66 | 53.66±1.69 | 53.71±2.73 | 54.73±2.97 | 49.29±3.14 | 54.05±2.68 | 53.74±2.63 | 60.03±2.43 | 61.39±2.68 | 53.50±4.51 | 58.08±2.18 | 48.10±3.09 | 48.32±2.29 | **66.80±1.11** |
| | β=10 | 54.62±2.75 | 54.43±1.52 | 54.39±2.59 | 55.56±2.94 | 55.75±2.31 | 54.78±2.53 | 54.93±2.40 | 60.94±2.47 | 61.94±2.90 | 55.98±3.68 | 58.29±2.13 | 50.31±2.86 | 51.02±2.09 | **67.59±1.16** |
| | β=1 | 64.05±3.71 | 64.01±2.57 | 64.12±2.60 | 66.04±3.35 | 64.73±2.17 | 64.67±3.87 | 64.77±2.78 | 66.74±2.53 | 67.06±2.86 | 59.88±3.81 | 61.39±2.08 | 51.76±3.57 | 54.55±4.05 | **73.61±1.72** |
| | β=0.1 | 84.30±7.56 | 82.25±5.62 | 80.41±6.57 | 84.33±5.76 | 84.25±5.23 | 84.43±5.57 | 76.49±8.11 | 81.29±5.37 | 82.97±5.24 | 75.31±2.91 | 78.18±3.03 | 61.37±3.99 | 59.17±6.14 | **87.52±4.03** |
| IMDB | Uniform | 44.40±2.91 | 44.22±2.80 | 44.46±2.77 | 44.79±2.53 | 39.42±4.42 | 44.27±3.04 | 44.33±3.11 | 44.94±3.42 | 44.73±3.50 | 42.38±5.75 | 43.43±2.56 | 37.42±1.65 | 36.46±1.20 | **48.67±2.37** |
| | β=10 | 46.61±2.51 | 46.44±2.59 | 46.30±2.20 | 47.53±3.11 | 48.09±7.32 | 46.62±2.51 | 46.66±2.36 | 47.36±2.32 | 47.32±2.33 | 46.54±4.91 | 47.71±2.26 | 39.86±2.64 | 39.27±1.94 | **51.78±1.66** |
| | β=1 | 55.91±6.96 | 55.92±5.93 | 56.35±6.43 | 60.19±4.57 | 51.13±4.83 | 56.37±6.52 | 56.47±5.26 | 57.91±5.39 | 57.05±4.28 | 48.62±5.42 | 53.26±2.66 | 53.46±5.89 | 53.55±6.89 | **60.26±3.60** |
| | β=0.1 | 60.02±6.95 | 60.19±5.57 | 60.28±4.99 | 62.32±4.89 | 58.01±5.44 | 60.01±4.70 | 59.85±4.63 | 60.15±5.48 | 60.85±4.96 | 51.75±3.13 | 57.41±2.11 | 62.68±7.65 | 62.18±6.60 | **64.42±4.75** |
| Yelp | Uniform | 68.91±3.99 | 70.30±2.24 | 70.44±3.09 | 73.24±2.59 | 54.54±4.42 | 70.37±3.50 | 71.08±3.20 | 66.36±10.02 | 69.43±3.44 | 63.23±3.92 | 65.48±2.61 | 72.92±2.64 | 73.58±2.56 | **75.37±1.64** |
| | β=10 | 70.48±3.06 | 72.49±2.88 | 70.67±3.84 | 74.85±3.34 | 57.47±3.47 | 71.28±2.83 | 71.49±2.19 | 61.40±9.89 | 64.13±4.18 | 64.72±4.12 | 67.71±2.39 | 74.83±2.51 | 74.69±2.45 | **77.10±2.07** |
| | β=1 | 77.86±5.16 | 78.18±5.93 | 77.40±5.80 | 80.01±4.86 | 72.58±6.78 | 79.64±4.97 | 79.86±4.54 | 67.29±9.41 | 70.48±5.39 | 68.62±5.23 | 71.23±2.24 | 77.52±5.22 | 79.36±5.50 | **82.25±3.61** |
| | β=0.1 | 87.29±6.08 | 88.01±3.39 | 88.89±4.32 | 87.62±3.38 | 84.63±5.03 | 88.22±2.30 | 87.85±3.82 | 80.26±7.91 | 83.71±5.48 | 72.29±4.99 | 73.89±2.37 | 85.46±5.55 | 88.29±4.83 | **91.16±2.99** |

**Table 3: Convergence round results for 4 datasets with different partitioning methods, showing the mean and standard deviation of convergence rounds, where a lower mean signifies faster convergence, a lower standard deviation reflects greater stability across repeated experiments.**

| Dataset | Partition | FedAvg | FedProx | MOON | SCAFFOLD | FedDyn | FedAW | FedWT | FedPROTO | FedTGP | FedGCN | FedSAGE | FedLit | FedHGN | MAFedHGL |
|---|---|---|---|---|---|---|---|---|---|---|---|---|---|---|---|
| ACM | Uniform | 180 ± 33 | 195 ± 33 | 174 ± 42 | 181 ± 35 | 184 ± 27 | 162 ± 44 | 172 ± 40 | 141 ± 50 | 138 ± 40 | 472 ± 62 | 553 ± 102 | 401 ± 32 | 389 ± 31 | **107 ± 31** |
| | β=10 | 140 ± 29 | 149 ± 32 | 154 ± 39 | 143 ± 32 | 156 ± 56 | 124 ± 22 | 133 ± 49 | 90 ± 21 | 107 ± 28 | 461 ± 38 | 523 ± 96 | 310 ± 40 | 295 ± 64 | **86 ± 20** |
| | β=1 | 147 ± 37 | 148 ± 28 | 161 ± 45 | 156 ± 34 | 161 ± 48 | 116 ± 27 | 131 ± 31 | 116 ± 42 | 138 ± 23 | 424 ± 67 | 546 ± 78 | 332 ± 56 | 316 ± 38 | **96 ± 21** |
| | β=0.1 | 152 ± 28 | 166 ± 30 | 179 ± 34 | 182 ± 29 | 258 ± 38 | 133 ± 28 | 183 ± 47 | 124 ± 35 | 134 ± 46 | 416 ± 31 | 528 ± 76 | 407 ± 68 | 256 ± 90 | **83 ± 26** |
| DBLP | Uniform | 254 ± 50 | 281 ± 32 | 248 ± 39 | 273 ± 13 | 198 ± 25 | 241 ± 38 | 253 ± 32 | 164 ± 22 | 187 ± 31 | 277 ± 72 | 379 ± 74 | 289 ± 63 | 277 ± 53 | **121 ± 18** |
| | β=10 | 207 ± 31 | 191 ± 38 | 189 ± 44 | 195 ± 25 | 187 ± 17 | 187 ± 25 | 178 ± 32 | 162 ± 48 | 170 ± 35 | 283 ± 68 | 367 ± 72 | 316 ± 27 | 310 ± 23 | **92 ± 14** |
| | β=1 | 147 ± 28 | 164 ± 47 | 140 ± 36 | 144 ± 37 | 186 ± 44 | 136 ± 42 | 140 ± 48 | 156 ± 32 | 192 ± 28 | 223 ± 52 | 298 ± 82 | 249 ± 43 | 252 ± 41 | **117 ± 18** |
| | β=0.1 | 139 ± 38 | 134 ± 39 | 135 ± 44 | 139 ± 19 | 173 ± 54 | 142 ± 28 | 133 ± 31 | 184 ± 25 | 176 ± 39 | 201 ± 57 | 328 ± 76 | 326 ± 94 | 251 ± 97 | **86 ± 22** |
| IMDB | Uniform | 114 ± 22 | 123 ± 22 | 113 ± 20 | 128 ± 28 | 127 ± 32 | 125 ± 24 | 113 ± 25 | 109 ± 21 | 118 ± 26 | 254 ± 68 | 289 ± 32 | 220 ± 20 | 224 ± 40 | **82 ± 18** |
| | β=10 | 135 ± 28 | 121 ± 24 | 127 ± 28 | 127 ± 22 | 138 ± 31 | 143 ± 25 | 157 ± 42 | 98 ± 32 | 102 ± 31 | 255 ± 63 | 292 ± 41 | 224 ± 23 | 223 ± 21 | **69 ± 11** |
| | β=1 | 146 ± 24 | 161 ± 28 | 154 ± 40 | 159 ± 27 | 143 ± 32 | 164 ± 31 | 97 ± 30 | 132 ± 29 | 134 ± 28 | 231 ± 37 | 276 ± 35 | 218 ± 20 | 212 ± 26 | **72 ± 22** |
| | β=0.1 | 144 ± 26 | 157 ± 33 | 168 ± 42 | 178 ± 33 | 156 ± 35 | 181 ± 42 | 96 ± 23 | 82 ± 33 | 89 ± 29 | 198 ± 41 | 263 ± 38 | 212 ± 74 | 183 ± 65 | **76 ± 24** |
| Yelp | Uniform | 269 ± 53 | 316 ± 90 | 246 ± 40 | 317 ± 103 | 252 ± 46 | 243 ± 31 | 231 ± 34 | 266 ± 22 | 262 ± 32 | 321 ± 42 | 489 ± 54 | 349 ± 60 | 370 ± 69 | **207 ± 36** |
| | β=10 | 299 ± 43 | 270 ± 66 | 265 ± 82 | 273 ± 40 | 234 ± 52 | 257 ± 47 | 223 ± 32 | 278 ± 36 | 235 ± 29 | 382 ± 67 | 483 ± 56 | 328 ± 56 | 377 ± 78 | **203 ± 30** |
| | β=1 | 256 ± 42 | 240 ± 62 | 219 ± 55 | 250 ± 75 | 184 ± 47 | 220 ± 24 | 169 ± 57 | 221 ± 43 | 187 ± 53 | 372 ± 56 | 429 ± 47 | 247 ± 54 | 287 ± 66 | **159 ± 49** |
| | β=0.1 | 258 ± 53 | 238 ± 62 | 207 ± 50 | 330 ± 86 | 267 ± 43 | 287 ± 64 | 150 ± 33 | 143 ± 43 | 162 ± 32 | 328 ± 39 | 419 ± 50 | 212 ± 41 | 188 ± 34 | **117 ± 42** |

edges, and edge types. A smaller value of $\alpha$ results in greater disparities in data distribution across clients. Unless otherwise specified, we set $K = 5$, data detail for clients are shown in Table 1.

**Baseline.** To validate the effectiveness of the proposed method, we designed several baseline algorithms for comparison, including FHGL algorithms such as FedHGN [6] and FedLit [43], FGL algorithms like FedSage [50] and FedGCN [46], and traditional FL algorithms that are effective for Non-IID data, such as FedAvg [27], FedProx [20], SCAFFOLD [16], MOON [19], FedDyn [1], FedAW [36], FedWT [36], FedPROTO [35], and FedTGP [49]. Among them, FGL baselines convert local HG data into homogeneous graphs for training, while FL algorithms designed for Euclidean data, like FedAvg, use HGNN for local learning.

**Implementation details.** Our implementation of HGNs is based on the *DGL* [38] library, while the processing of HG data relies on the *OpenHGNN* [15] library. GPU acceleration is performed on NVIDIA RTX 2080 TI. In each set of experiments, 10 repetitions are conducted, and the convergence criterion for accuracy and loss is set to 10 rounds without change. To ensure the fairness of the experiments, all methods use the same learning rate of 0.001 and the same number of local training epochs set to 10. The hidden embedding dimension of HGNN 64, the number of HGNN layers is set to 2. Both $\gamma_{\mathcal{E}}$ and $\gamma_{\mathcal{V}}$ for MAFedHGL are set to 1.

## 5.2 Overall Performance

Both Table 2 and Table 3 demonstrate that MAFedHGL outperforms all baseline methods on performance and convergence speed, exhibiting strong robustness to variations in datasets and partitioning methods. This indicates that MAFedHGL can capture complex correlations between multi-hop and multi-path nodes, strengthening the model's robustness to HG structures across different clients. By leveraging HGNN to infer missing semantic information and effectively integrating semantic patterns from different clients, it ensures global semantic consistency. MAFedHGL evaluates the connectivity between each client's nodes and their neighbors, assigning higher weights to clients with more complex structural information. This avoids the negative impact of structural discrepancies between clients on the global model, enabling the model to dynamically adapt to varying data distributions and enhancing its robustness and generalization ability. In HG scenarios, clients' graph structures can differ significantly, including variations in node and edge types and quantities. Due to HGNN's sensitivity to graph structure, traditional FL methods and federated graph learning struggle to capture these local structural differences, leading to poor generalization of the global model. Consequently, these methods may perform well in some cases but poorly in others. Both FedLit and FedHGN exhibit strong capabilities in extracting topological features, particularly in scenarios with smaller $\beta$, where HFHGL approaches become more

akin to vertical federated heterogeneous graph learning. However, it was also observed that these methods perform poorly when dealing with the DBLP dataset, which has a lower clustering coefficient.

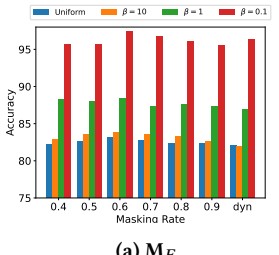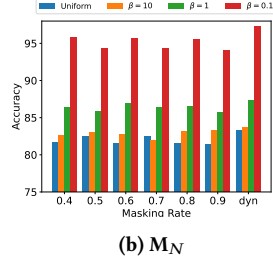

**(a) $M_E$**

**(b) $M_N$**

**Figure 2: Performance comparison under various masking rate on ACM dataset.**

## 5.3 Parameter Analysis

MAFedHGL forces the model to learn latent features by masking the adjacency matrix and node features, with $M_E$ and $M_V$ representing the corresponding masking ratios. We conduct experiments to determine the optimal masking ratios, and the results on the ACM dataset are shown in Figure 2. It can be observed that the optimal value of $M_E$ under different data partitions is consistently around 0.5, whereas $M_V$ performs better with dynamic partitioning, and there is no single value that achieves optimal results across all partitions. The same pattern was observed in other datasets as well.

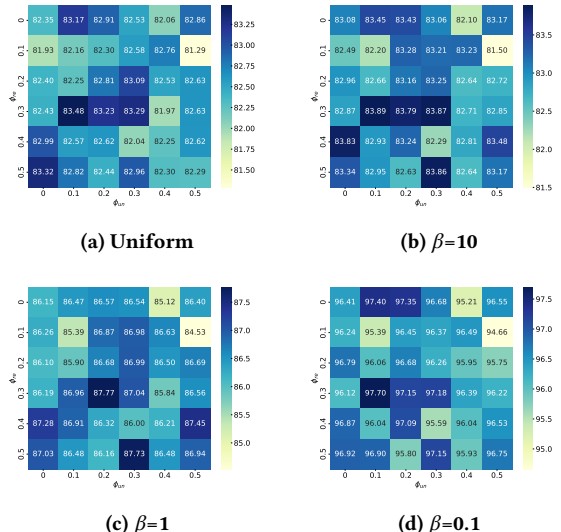

**(a) Uniform**

**(b) $\beta$=10**

**(c) $\beta$=1**

**(d) $\beta$=0.1**

**Figure 3: Performance comparison under various $\phi_{un}$ and $\phi_{re}$ on ACM dataset.**

We also conduct parameter experiments on $\phi_{re}$ and $\phi_{un}$ to investigate whether node feature perturbation should be retained or replaced with features from other nodes. The results on the

ACM dataset are shown in Figure 3, where we find that $\phi_{re}$ and $\phi_{un}$ exhibit similar distribution trends under different partitions. Therefore, for each dataset, we determine the values of these two parameters through tuning.

**Table 4: Results of different model variants.**

| Dataset | Partition | w/o HTM | w/o SCM | w/o CCA | MAFedHGL |
|---------|-----------|---------|---------|---------|----------|
| ACM | Uniform | 81.03±2.80 | 81.96±1.39 | 82.16±1.85 | **83.30±1.39** |
| | $\beta$=10 | 81.15±1.67 | 82.05±2.13 | 82.46±1.23 | **83.67±0.52** |
| | $\beta$=1 | 85.45±2.61 | 86.10±1.20 | 86.42±1.70 | **87.33±1.76** |
| | $\beta$=0.1 | 95.30±2.18 | 96.19±1.46 | 96.58±1.76 | **97.36±1.70** |
| DBLP | Uniform | 64.79±1.79 | 64.84±1.28 | 65.61±1.21 | **66.83±1.17** |
| | $\beta$=10 | 65.38±2.28 | 67.34±1.20 | 67.08±1.95 | **67.68±1.19** |
| | $\beta$=1 | 71.45±2.02 | 72.42±1.97 | 72.62±1.92 | **73.66±1.74** |
| | $\beta$=0.1 | 85.94±5.29 | 86.51±4.33 | 86.83±4.78 | **87.56±4.09** |
| IMDB | Uniform | 47.14±2.54 | 47.47±2.11 | 48.14±2.37 | **48.74±2.48** |
| | $\beta$=10 | 49.63±2.60 | 50.52±1.55 | 50.63±2.08 | **51.86±1.75** |
| | $\beta$=1 | 57.95±5.13 | 60.08±4.96 | 59.61±4.35 | **60.31±3.66** |
| | $\beta$=0.1 | 62.16±6.12 | 62.50±6.18 | 63.50±5.55 | **64.49±4.76** |
| Yelp | Uniform | 73.09±2.15 | 73.55±2.27 | 74.11±1.67 | **75.40±1.70** |
| | $\beta$=10 | 74.37±3.25 | 75.71±2.85 | 76.00±2.59 | **77.14±2.12** |
| | $\beta$=1 | 80.00±4.74 | 79.83±4.72 | 80.79±4.60 | **82.29±3.62** |
| | $\beta$=0.1 | 88.32±3.34 | 88.71±2.91 | 89.78±3.14 | **91.17±3.08** |

## 5.4 Ablation Study

MAFedHGL employs three main strategies: High-order Topology Mining (HTM), Semantic Consistency Modeling (SCM), and Clustering Coefficient Aggregation (SSA). We perform ablation experiments to evaluate the contribution of each strategy to the model. The results, shown in Table 4, indicate that removing any of the three strategies leads to a performance drop, confirming their effectiveness. Among them, the removal of HTM causes the most significant performance decline, while removing SSA results in the least decrease in stability across repeated experiments.

## 6 CONCLUSION

In this paper, we first explore the emerging field of horizontal federated heterogeneous graph learning, addressing the challenges posed by distributed heterogeneous data. We propose a novel method, Multi-scale Adaptive Horizontal Federated Heterogeneous Graph Learning MAFedHGL, aimed at enhancing model robustness and performance in the face of imbalanced client data distributions and significant structural differences in graphs. By employing high-order topology mining to uncover latent relationships and dynamic semantic consistency modeling to address missing semantic connections, we introduc a clustering coefficient-based aggregation strategy, which assigns weights to clients based on the tightness of their node-to-neighbor connections. MAFedHGL effectively improves feature representation and global understanding across participants. Experimental results on multiple public heterogeneous graph datasets demonstrated that MAFedHGL consistently achieves high performance while maintaining robustness across diverse data scenarios, highlighting its potential for practical applications in privacy-sensitive domains.

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
