# OpenReview forum: "Horizontal Federated Heterogeneous Graph Learning: A Multi-Scale Adaptive Solution to Data Distribution Challenges"
_ACM.org/TheWebConf/2025/Conference — WWW 2025 Poster_

### Official Review · Reviewer_bduA · 2024-12-01

**Novelty:** 6
**Technical Quality:** 6

**Review:**

This paper proposed MAFedHGL to overcome imbalanced client data distributions and significant structural differences in heterogeneous graph learning. The main idea of MAFedHGL is masking some graph structures and node attributes during the learning process, enforcing the model to master its ability to infer missing structures and retain global and local semantic consistency. Lastly,  a clustering coefficient-based aggregation strategy is introduced, which assigns weights to clients based on the tightness of their node-to-neighbor connections. As a result, MAFedHGL effectively improves feature representation and global understanding across clients.

Specifically,

1. The problem is well motivated and formulated. Three technical challenges are identified and well-studied. They are structural sensitivity, semantic complexity, and personalized federated aggregation.

2. The proposed MAFedHGL is technically sound, solving three identified challenges reasonably.

3. Experiments are solid and the results show that the proposed MAFedHGL is highly effective, outperforming the selected 13 baselines significantly.

4. The paper is well written. I very enjoy reading this paper. In general, I think this paper is of high quality although the studied problem, i.e., horizontal federated graph learning, is not totally new.

**Questions:**

Is the clustering coefficient $C_k$ possible to lead to privacy leakage? Why?

**Reviewer Confidence:**

3: The reviewer is confident but not certain that the evaluation is correct

**Scope:**

4: The work is relevant to the Web and to the track, and is of broad interest to the community

---

### Official Review · Reviewer_FR6Y · 2024-12-02

**Novelty:** 5
**Technical Quality:** 5

**Review:**

This paper proposes a novel multi-scale adaptive horizontal federated heterogeneous graph learning method, which uses a random masking
 mechanism to force the model to infer missing connections. However, there are still some issues that need further clarification:

1. In Section 4.4, the authors claim that this work is privacy-preserving, however, it is not the case that uploading only the parameters is truly privacy-preserving. A large number of studies have shown that reverse engineering through the gradient of the model can still recover data distribution and other information to a large extent, and this problem has not been well studied in this paper.

2. The necessary ablation analysis experiments are lacking. The role of the components of the proposed method has not been demonstrated experimentally.

3. The cost of computation and communication is also an important problem in federated learning, but the lack of research on these aspects in this paper makes the evaluation of methods incomplete.

4. Federated graph learning is also a challenge in terms of efficiency. This paper does not analyze the time and space complexity of this method, nor does it compare the efficiency from an experimental point of view with baseline models.

5. This paper lacks a convergence analysis of the algorithm. In the field of Federated Learning, theoretical and experimental convergence analysis has always been a necessity and should be considered to be provided in the supplementary material even if it is not provided in the main text.

**Questions:**

See Review.

**Reviewer Confidence:**

3: The reviewer is confident but not certain that the evaluation is correct

**Scope:**

3: The work is somewhat relevant to the Web and to the track, and is of narrow interest to a sub-community

---

### Official Review · Reviewer_sdoM · 2024-12-03

**Novelty:** 6
**Technical Quality:** 5

**Review:**

For the problem of data from different parties often differ in topology and semantic distribution, this paper propose a multi-scale adaptive horizontal federated heterogeneous graph learning method. This model contains three parts, i.e., A random masking mechanism forces the model to infer missing connections, dynamic semantic consistency modeling uses a masking matrix to recover and integrate diverse node attributes and clustering coefficients as aggregation weights enables clients with richer structural information to contribute more effectively to the global model.

Comments:
The article has a clear structure and is easy to read. The illustrations are meticulously and beautifully designed, the workload is substantial, and the experimental design is comprehensive.

Weakness:

1. The code has not been made publicly available.
2. The motivation is not strong enough. Lines 103-140 primarily discuss the challenges addressed by the paper. While this section is detailed, it is overly lengthy and fails to highlight the key points effectively. The discussion of these challenges is largely based on a literature review, lacking illustrative examples or experimental evidence. Additionally, some arguments are unsupported. For instance, in lines 132-134, the claim that "average aggregation strategies fail to distinguish between the quality and structural complexity of data from different clients" lacks references and supporting evidence, making it unconvincing. It is suggested that the authors refine this section in future revisions, distilling the innovative points more clearly.
3.  The solutions proposed for the challenges fail to convincingly address the issues and seem to avoid the critical points. For example, in response to the first challenge, "magnifies the structural sensitivity of HGNN," the paper proposes the solution: Through High-order Topology Mining (HTM), the model captures multi-hop and multi-path connections between nodes, enabling comprehensive analysis of multi-scale features in complex graph structures. However, I do not see a clear correspondence between this solution and the stated challenge. It is recommended that future versions provide intuitive solutions to each challenge, addressing them one by one. This approach would make the overall solution more convincing and coherent.

**Questions:**

See above.

**Reviewer Confidence:**

2: The reviewer is willing to defend the evaluation, but it is likely that the reviewer did not understand parts of the paper

**Scope:**

3: The work is somewhat relevant to the Web and to the track, and is of narrow interest to a sub-community

---

### Official Review · Reviewer_Ud4a · 2024-12-03

**Novelty:** 5
**Technical Quality:** 5

**Review:**

The paper develops a horizontal federated heterogeneous graph learning method, which is, according to the authors, the first work that explores heterogeneous graph data processing in federated learning. The research topic exhibits certain interesting, and there lacks works that discuss this issue.

The main idea of the work is to employ a missing attribute reconstruction module and an edge prediction module to explore the high-order topology data structure and consistent semantic information, with the meta-path based heterogeneous graph neural network as the underlying model. The overall framework indicates certain novelty. Also, several experiments are performed.

**Questions:**

The main concerns are as follows:
1. There are some typos to be polished. For the ablation study, the table includes "CCA", which might be "SSA" according to the paper.
2. According to \textbf{Data Partitioning}, the uniform strategy seems to refer to iid condition, but all the methods achieve the lowest performance under this condition. Could the authors explain the potential reasons?
3. The ablation study indicates each single module does not contribute too much to performance improvement. Without any module, the model performance is still comparable to the comparison methods. It is recommended to analyze the potential reasons, and a more comprehensive ablation study might be needed.

**Reviewer Confidence:**

3: The reviewer is confident but not certain that the evaluation is correct

**Scope:**

4: The work is relevant to the Web and to the track, and is of broad interest to the community